# P2X7 Receptors and TMEM16 Channels Are Functionally Coupled with Implications for Macropore Formation and Current Facilitation

**DOI:** 10.3390/ijms22126542

**Published:** 2021-06-18

**Authors:** Kate Dunning, Adeline Martz, Francisco Andrés Peralta, Federico Cevoli, Eric Boué-Grabot, Vincent Compan, Fanny Gautherat, Patrick Wolf, Thierry Chataigneau, Thomas Grutter

**Affiliations:** 1CNM Team, Centre National de la Recherche Scientifique, CAMB UMR 7199, Faculté de Pharmacie, Université de Strasbourg, 67401 Illkirch, France; katelouisedunning@gmail.com (K.D.); adeline.martz@unistra.fr (A.M.); fperalta@unistra.fr (F.A.P.); fcevoli@unistra.fr (F.C.); fanny.gautherat@etu.unistra.fr (F.G.); patrick.wolf@etu.unistra.fr (P.W.); thierry.chataigneau@unistra.fr (T.C.); 2Institute for Advanced Studies (USIAS), University of Strasbourg, 67000 Strasbourg, France; 3Institut des Maladies Neurodégénératives, Université de Bordeaux, UMR 5293, 33000 Bordeaux, France; eric.boue-grabot@u-bordeaux.fr; 4Institut des Maladies Neurodégénératives, CNRS, UMR 5293, 33000 Bordeaux, France; 5Institut de Génomique Fonctionnelle (IGF), Université de Montpellier, CNRS, INSERM, 34094 Montpellier, France; vincent.compan@igf.cnrs.fr; 6Laboratoire d’Excellence Canaux Ioniques d’Intérêt Thérapeutique (LabEx ICST), 34094 Montpellier, France; 7Centre National de la Recherche Scientifique, Institut des Neurosciences Cellulaires et Intégratives, UPR-3212, Université de Strasbourg, 67000 Strasbourg, France

**Keywords:** P2X7, anoctamin, ATP sensitization, cell permeabilization, purinergic receptor, ion channel

## Abstract

P2X7 receptors (P2X7) are cationic channels involved in many diseases. Following their activation by extracellular ATP, distinct signaling pathways are triggered, which lead to various physiological responses such as the secretion of pro-inflammatory cytokines or the modulation of cell death. P2X7 also exhibit unique behaviors, such as “macropore” formation, which corresponds to enhanced large molecule cell membrane permeability and current facilitation, which is caused by prolonged activation. These two phenomena have often been confounded but, thus far, no clear mechanisms have been resolved. Here, by combining different approaches including whole-cell and single-channel recordings, pharmacological and biochemical assays, CRISPR/Cas9 technology and cell imaging, we provide evidence that current facilitation and macropore formation involve functional complexes comprised of P2X7 and TMEM16, a family of Ca^2+^-activated ion channel/scramblases. We found that current facilitation results in an increase of functional complex-embedded P2X7 open probability, a result that is recapitulated by plasma membrane cholesterol depletion. We further show that macropore formation entails two distinct large molecule permeation components, one of which requires functional complexes featuring TMEM16F subtype, the other likely being direct permeation through the P2X7 pore itself. Such functional complexes can be considered to represent a regulatory hub that may orchestrate distinct P2X7 functionalities.

## 1. Introduction

Purinergic P2X7 receptors (P2X7) are members of the P2X receptor family, a class of membrane proteins forming trimeric channels activated by extracellular ATP [1]. They are expressed in different cell types, mainly in immune and glial cells. Following ATP gating, P2X7 initiates nonselective metal cation flux (Na^+^, K^+^ and Ca^2+^), which in turn triggers distinct activation pathways such as the secretion of pro-inflammatory cytokines or modulation of cell death. Owing to its key role in multiple pathologies, including chronic inflammation, neurodegeneration, neuropathic pain, metabolic diseases, rheumatoid arthritis, Crohn’s disease and cancer, P2X7 has become a particularly relevant therapeutic target, sparking intense interest for drug development [2].

Among the P2X family, P2X7 exhibits several hallmark features, including an elongated intracellular C-terminus, the necessity of unusually high concentrations of ATP for channel activation, and ATP-evoked current facilitation (also known as ATP sensitization) [3]. This process is caused by repeated or prolonged ATP applications, which yield a leftward shift of agonist affinity [4,5], and is inhibited by cholesterol [6,7], an essential component of the cell membrane. Recent work has further indicated that this inhibition may be counteracted by the presence of palmitoylated cysteine residues located in the intracellular C-terminus of the P2X7, suggesting that lipid composition may represent a powerful means to allosterically regulate receptor function [8].

In addition to ATP-evoked current facilitation, “macropore” formation or cell permeabilization is another hallmark of P2X7 [3,9]. First observed over forty years ago in mast cells [10], this process was initially attributed to the “P2Z” receptor before being reclassified as the seventh member of the P2X family, P2X7, following its first cloning [3]. Later confirmed in various other cell lines, including mouse RAW264 and human macrophages, microglia and HEK293 cells, across both recombinant and endogenous systems [3,11,12,13,14,15,16], this phenomenon allows the passage of nanometer-sized molecules (<900 Da) across the cell membrane. Macropore formation is usually monitored by the cellular uptake of fluorescent dyes [17], such as ethidium and YO-PRO-1, and the activation of this pathway eventually leads to cell death [9]. Often confounded with the ATP-evoked current facilitation phenomenon [18,19], several hypotheses have been proposed to explain macropore formation, including possible interactions with the pore forming pannexin hemichannel [20], but none have proved entirely satisfactory. One hypothesis that has remained prevalent until very recently was the so-called “pore dilation” theory [21,22], which postulated that ATP stimulation induces a progressive, time-dependent expansion of the pore, gradually increasing both current and large molecule permeability. This idea was appealing in its simplicity, but has been challenged by recent work [23], and a new permeation mechanism supported by biochemical [8], electrophysiological [23,24,25,26], photochemical [26] and modeling [23,26] data has emerged, in which the passage of small cationic molecules, such as *N*-methyl-d-glucamine (NMDG^+^) and spermidine, occurs immediately after ATP gating, via a non-dilated, open channel state. The main difference resides in the permeation rates, whereby NMDG^+^ ions flow at a reduced rate compared to that of Na^+^ [26].

At first glance, this new permeation mechanism appears to reconcile most of the data [27], but there remains an unanswered question as to how nanometer-sized molecules larger than NMDG^+^ (195 Da), such as YO-PRO-1 (376 Da) or even larger dyes (e.g., YOYO-1, 763 Da) [14], can directly permeate the P2X7 channel itself [16,28]. While reconstruction of an engineered P2X7, truncated at the N- and C-termini, into artificial liposomes supported the hypothesis that the pore itself is capable of YO-PRO-1 dye uptake, recent cryoelectron microscopy (cryo-EM) structures of ATP-bound, full-length rat P2X7 revealed that the open channel diameter measures only 5 Å, with no evidence of a dilated state [29], an aperture that seems too narrow to allow the passage of nanometer-sized molecules. Given that the minimal cross-section of YO-PRO-1 is ~7 Å, it is therefore hard to conceive that cellular uptake of molecules similar to or larger than YO-PRO-1 occurs solely through the P2X7 open pore itself, suggesting that additional routes are possible [16]. Given the importance of P2X7 in numerous pathologies, there is an urgent need to understand the molecular foundations of macropore formation and current facilitation, which undoubtedly remain two of the most enigmatic P2X features.

Recently, Ca^2+^-activated Cl^−^ channels (CaCCs) have been suggested to constitute one such possible route. Pharmacological inhibition of these channels prevents dye uptake in different cell lines, including HEK293 cells [30], human and mouse macrophages [14,30] as well as human microglia [13]. In addition, a functional coupling between these channels and P2X7 has also been described in *Xenopus* oocytes [31]. In mouse macrophages, the transmembrane protein 16F (TMEM16F or Anoctamin-6) was identified as one of these CaCCs and proposed to mediate effects essential for innate immunity downstream of P2X7 [30]. TMEM16F belongs to the TMEM16 family, a vast group of membrane proteins characterized by a high degree of functional diversity, including roles such as Ca^2+^-activated Cl^−^ channels, Ca^2+^-dependent phospholipid scramblases, and dual function non-selective ion channel/phospholipid scramblases [32,33,34]. There are ten TMEM16 proteins (TMEM16A-K, excluding I) found in humans, and recent evidence suggests that TMEM16F functions as a dual Ca^2+^-dependent phospholipid scramblase and Ca^2+^-activated poorly selective (rather than purely Cl^−^) ion channel. The scramblase activity of TMEM16F, which is its primary function, led to the suggestion that its activation by ATP-evoked P2X7 Ca^2+^ flux mediates downstream intracellular signaling processes, resulting in outcomes such as membrane blebbing and apoptosis [30]. However, the significance of TMEM16F to P2X7 features, such as macropore formation and current facilitation, remains to be elucidated.

In this work, we set out to test the hypothesis that P2X7 and TMEM16 form functional complexes in HEK293T cells and *Xenopus* oocytes. We combined mutagenesis techniques, genetic approaches, biochemical experiments, single-channel and whole-cell recordings, and cell imaging to show that these complexes have implications in both agonist-evoked current facilitation and macropore formation. Our data thus offer new insights into these enigmatic processes that have proved difficult to explain for 40 years.

## 2. Results

### 2.1. ATP-Evoked Current Facilitation Results in Increased P2X7 Open Probability

To investigate the mechanism of current facilitation at the molecular level, we used patch clamp electrophysiology to record single-channel activity from outside-out patches of HEK293T cells transiently transfected with rat P2X7 (rP2X7). To the best of our knowledge, no such single-channel currents have been previously described in the literature. To avoid non-specific channel activation evoked by millimolar concentrations of ATP [35], we used the more potent P2X7 agonist 2′(3′)-*O*-(4-benzoylbenzoyl)ATP (BzATP), which activates the channel in the low μM range. We further reduced the amount of extracellular divalent cation concentrations to dampen P2X7 channel inhibition [36]. In control patches, barely visible BzATP-gated single-channel openings and closings were detected when data were filtered at a corner frequency (f_c_) of 1 kHz (Appendix A). However, by further filtering data at 100 Hz, clearly resolved unitary currents were observed (Figure 1A, left) that rapidly disappeared upon BzATP washout (Appendix A). Under such conditions, the dead time (t_d_) was equal to 1.8 ms, and consequently transitions shorter than t_d_ were ignored. These unitary currents can be attributed to rP2X7 as no currents were recorded in the absence of BzATP, in the presence of the highly selective P2X7 antagonist AZ10606120 co-applied with BzATP, nor in non-transfected cells stimulated with BzATP (Figure 1A, right, and Appendix A).

Two unitary current amplitudes of 1.89 ± 0.26 pS and 3.39 ± 0.44 pS (*n* = 7 patches, Figure 1D, top) were observed, in agreement with previous data obtained with ATP from the human P2X7 receptor (hP2X7) expressed in oocytes [37], although the amplitude of these currents was 4- to 5-fold lower than that of the human counterpart. Given that we analyzed only patches containing three or fewer channels, these currents likely did not originate from stackings of double openings, because the open channel probability (N*P*o) of the highest amplitude current was higher than that of the lowest amplitude current (0.14 for the highest vs. 0.03 for the lowest amplitude), a situation that does not follow a binomial distribution (see Section 4). The simultaneous presence of these two discrete conductance levels, which we defined as O1 and O2, was observed in 5 out of 7 patches (two patches exhibited only the O2 state) and most likely reflects a dual gating mechanism, as suggested previously for hP2X7 [37]. Analysis of open and shut-time histograms fitted with multiple exponentials revealed a mean open time for O1 state (τ_O1_) of 24 ms and mean open time for O2 state (τ_O2_) of 44 ms, and a mean closed time (τ_C_) of 185 ms (Figure 1E, top and Appendix A, 1518 events).

Next, we devised a protocol in which current facilitation was first induced by a 30 s perfusion of BzATP before establishing an excised membrane patch in the outside-out configuration (Figure 1B). We first verified that such long agonist perfusion induced a strong whole-cell current facilitation that is similar to eight repeated short agonist applications (Appendix A), as previously described [6]. We observed some variability in the degree of whole-cell current facilitation and a significant increase in the deactivation time constant of facilitated currents during BzATP washout (Appendix A), two features that were previously reported [6,38].

In patches having undergone a 30 s BzATP facilitation treatment, the two unitary current amplitudes were still observed (in 5 out of 6 patches), and conductance values were similar to those of untreated conditions (*p* = 0.279 for O1 and 0.383 for O2, unpaired Student’s *t*-test, Figure 1C,D, middle, *n* = 6 patches). However, a dramatic increase of the mean open times of both conductance levels was recorded (~10-fold for O1, τ_O1_ = 234 ms, and ~5-fold for O2, τ_O2_ = 211 ms), along with a modest increase of the mean closed time (~1.5-fold, τ_C_ = 279 ms, Figure 1E, middle and Appendix A, 754 events). As a result, N*P*o for both conductance states combined increased by ~2.9-fold, from 0.17 to 0.47. Of note is that this increase is not only in the same range as BzATP-evoked current facilitation observed in the whole-cell configuration (between ~3- to ~4-fold), but also consistent with the increased deactivation time constant after facilitation. This suggests that in HEK293T cells, agonist-evoked current facilitation predominantly originates from a large increase of rP2X7 open probability, but not of single-channel conductance, further disfavoring the pore dilation theory.

### 2.2. Plasma Membrane Cholesterol Depletion Increases P2X7 Open Probability

To uncover the molecular basis of current facilitation, we recorded BzATP-gated unitary currents in excised outside-out patches in which membrane cholesterol was acutely depleted by methyl-β-cyclodextrin (MCD), a cholesterol chelating agent (Figure 1B). It has been shown that MCD potentiates P2X7 function, by dramatically enhancing whole-cell currents evoked by the first agonist application [6]. As a result, initial currents recorded from MCD-treated cells are of significantly greater magnitude than those recorded in control cells, showing instead similar magnitude to control cell currents recorded after BzATP facilitation. After verifying that this was effectively the case in the whole-cell configuration (Appendix A), we recorded single-channel currents from outside-out patches excised from cells that were pre-treated with MCD. Similarly to BzATP-evoked current facilitation, we found that MCD treatment enhanced mean open times of both conductance states when compared to control patches (~16-fold for O1, τ_O1_ = 378 ms and ~3-fold for O2, τ_O2_ = 134 ms, 396 events), with a small but statistically insignificant reduction of unitary current amplitudes (*p* = 0.146 for O1 and 0.080 for O2, unpaired Student’s *t*-test, Figure 1C–E, bottom; all patches contained both O1 and O2 states, except two patches that exhibited only the O2 state, *n* = 7 patches). In line with deactivation kinetics of facilitated currents, MCD treatment also significantly increased the deactivation time constant of BzATP-evoked currents, an effect observed from the first application (Appendix A). These data demonstrate that MCD treatment recapitulates ATP-evoked current facilitation, presumably by dissociating cholesterol from the receptor.

Taken together, these results show that prolonged activation of P2X7 enhances open probability, an effect that is recapitulated by depletion of plasma membrane cholesterol.

### 2.3. A Functional Interplay between P2X7 and TMEM16

Having shown that current facilitation results in an increase of the single-channel open probability, we next asked whether this increase is related to macropore formation. Because CaCC inhibitors have been previously shown to inhibit P2X7-dependent dye uptake [14,30], we tested whether they are able to inhibit P2X7 currents which have been facilitated by four repeated applications of BzATP. We focused on tannic acid (TA), flufenamic acid (FFA) and 9-anthracene-carboxylic acid (9-AC), for which functional modulation of TMEM16 activity, including TMEM16A [39,40], TMEM16B [41] and TMEM16F [30], has been recently shown in recombinant systems.

We found that the application of 20 μM TA alone for 60 s (referred to as “application alone”) to facilitated receptors abolished subsequent BzATP-evoked whole-cell currents in rP2X7-expressing HEK293T cells, while 1 mM 9-AC reduced subsequent currents by ~35% (Figure 2A,B). For FFA (100 μM), no inhibition was found in these conditions; however, a single co-application of FFA and BzATP at the end of FFA perfusion reduced subsequent BzATP-evoked currents to ~35% (Figure 2A,B). Contrary to TA and 9-AC, FFA inhibitions were fully reversible upon washout. Importantly, no inward current was induced by application of TA, 9-AC, or FFA alone (Appendix A). Thus, these data suggest that facilitated P2X7 currents are fully inhibited by TA, and partially inhibited by 9-AC and FFA.

To exclude a possible direct modulation of P2X7 activity by selected CaCC inhibitors, we investigated the effects of these inhibitors in Axolotl oocytes (*Ambystoma mexicanum*), which are reported to be electrophysiologically void of CaCCs [42,43]. Using two-electrode voltage-clamp (TEVC) and the “application alone” protocol, no P2X7 current inhibition was detected by 9-AC or FFA (Figure 2C,D), while about 50% inhibition was observed for both inhibitors in *Xenopus* oocytes (Figure 2E,F), which do express endogenous CaCCs [42]. Note that for FFA, inhibition took place using the “application alone” protocol directly, suggesting that washout dissociates bound FFA more slowly in *Xenopus* oocytes than in HEK293T cells. Current facilitation was observed in *Xenopus* oocytes, albeit not systematically, but not in Axolotl oocytes, suggesting that the absence of current facilitation is not correlated to an absence of CaCC inhibition. These data, therefore, strongly suggest that 9-AC and FFA do not act as inhibitors of rP2X7 itself, but instead that their partial inhibition of facilitated rP2X7 currents in HEK293T cells and *Xenopus* oocytes occurs through a functional interplay between rP2X7 and endogenous CaCC-sensitive channels such as human (h) hTMEM16.

In contrast to 9-AC and FFA, TA abolished currents in Axolotl oocytes (Figure 2C,D), as well as in *Xenopus* oocytes (Figure 2E,F), suggesting that it may directly inhibit the P2X7 channel itself through specific or non-specific mechanisms. The action of TA is therefore unclear, but we have decided to include the results throughout this manuscript to further document its action on P2X7.

### 2.4. The Functional Interaction of P2X7 and TMEM16 Contributes to Current Facilitation

Having shown the action of CaCC inhibitors on P2X7 facilitated currents, we next asked whether these inhibitors applied after the very first BzATP application in naïve cells prevent current facilitation. Inhibition was monitored by comparing currents evoked by repeated BzATP applications in the presence or absence of CaCC inhibitors. We found that TA and 9-AC not only strongly inhibited BzATP-evoked currents (measured at the fourth application), but also prevented current facilitation (Figure 3A,B). FFA was not different from the control, and therefore no inhibition was detected (Figure 3B). Recalling that 9-AC is not acting on P2X7 itself, these data suggest that the functional interplay between P2X7 and TMEM16 contributes to current facilitation.

### 2.5. The Functional Interaction of P2X7 and TMEM16 Is Retained in Excised Patches

Next, we asked whether inhibitions observed in the whole-cell configuration are retained in excised patches. Co-applications of BzATP and each of the CaCC inhibitors to rP2X7-expressing outside-out patches inhibited BzATP-evoked single-channel currents to levels (Figure 3C,D) that were similar to those obtained in the whole-cell configuration when inhibitors were applied before facilitation (Figure 3A,B). The only exception was FFA, for which no or very weak inhibition was observed in the whole-cell configuration while substantial inhibition was detected in excised patches (Figure 3D and Appendix A). The reason for this is unclear but may be related to differences in the access of FFA to its binding site between whole-cell and single-channel conditions. Single-channel inhibitions were reversible upon washout, even for TA, and levels of inhibition were similar to those obtained by co-applying the selective P2X7 inhibitor AZ10606120 with BzATP. Overall, 9-AC inhibition data indicate that the interplay between rP2X7 and hTMEM16 remains stable in excised patches, and therefore does not require an intact cell and associated cellular components.

Noticeably, 9-AC and FFA inhibitors largely decreased the frequency at which we observed BzATP-evoked single-channel openings, but not the unitary conductance of those few currents which remained (Figure 3C), thereby further suggesting that inhibition produced by these compounds does not occur on the P2X7 pore itself via a fast open-channel block mechanism, but rather through an allosteric inhibition mechanism that modifies P2X7 gating transitions.

### 2.6. The Functional Interaction of P2X7 and TMEM16 Contributes Partially to YO-PRO-1 Uptake

Having shown the existence of functional interaction between rP2X7 and hTMEM16, we next asked whether this interaction is involved in dye uptake in HEK293T cells. We used YO-PRO-1, a fluorescent dye of 376 Da, and showed robust dye uptake in cells expressing rP2X7 following 10 μM BzATP application (Figure 4A,B).

No dye uptake was observed in the absence of BzATP or in its presence in non-transfected cells (Figure 4C). TA co-applied with BzATP abolished dye uptake, but curiously, the extent of inhibition was greater than that of controls (i.e., without BzATP or in non-transfected cells) (Figure 4A–C). This suggests that TA, in addition to its dye uptake inhibition, may quench YO-PRO-1 fluorescence before dye uptake. To test this hypothesis, we acquired emission spectra of YO-PRO-1 bound to double strand DNA (dsDNA) in the absence or presence of TA and found that addition of TA led to immediate and complete quenching of fluorescence intensity (Figure 4D). In contrast, no fluorescence quenching was observed with 9-AC or FFA (Figure 4D), even 15 min after inhibitor addition. These data demonstrate that TA is a powerful YO-PRO-1 quencher, and therefore should not be used as a pharmacological inhibitor in dye uptake assays.

Next, we found that 9-AC significantly reduced BzATP-stimulated YO-PRO-1 uptake, by ~40% while no dye uptake inhibition was observed with FFA (Figure 4A–C). These results suggest that the functional interaction between P2X7 and TMEM16 contributes partially to dye uptake. Because TMEM16F was previously shown to mediate functional effects downstream of P2X7 [30], we tested the hypothesis that the functional interaction between P2X7 and TMEM16 features the hTMEM16F subtype.

### 2.7. TMEM16F Underlies a Component of YO-PRO-1 Uptake

To investigate the contribution of hTMEM16F to dye uptake, we applied CRISPR/Cas9 gene editing to produce a hTMEM16F-deficient (16F-null) HEK293T cell line (Appendix A). We verified the absence of endogenously expressed hTMEM16F in this cell line compared to wild-type (WT) HEK293T cells by Western blot (Figure 5A). Before addressing dye uptake, we first tested whether rP2X7 expression is influenced by endogenous hTMEM16F, by comparing rP2X7 subunit expression in both HEK293T and 16F-null cell lines. We used a rP2X7 construct that was tagged at its C-terminus with a myc sequence and found a significant increase of both total and cell surface expression of rP2X7 subunit in 16F-null cell lines compared to HEK293T cells (Figure 5B,C). These data reveal that the presence of endogenous hTMEM16F channels reduces rP2X7 subunit expression when this latter is ectopically expressed in HEK293T cells.

Next, we measured BzATP-induced dye uptake in the rP2X7-expressing 16F-null cell line. A significant ~40% reduction of YO-PRO-1 uptake was detected compared to HEK293T cells, while controls in the absence of BzATP showed no dye uptake (Figure 5D–G). These data thus unambiguously demonstrate a statistically significant contribution of hTMEM16F to dye uptake.

Nevertheless, dye uptake was not abolished in 16F-null cells, raising the possibility that the remaining dye uptake involves other TMEM16 members and/or intrinsic passage through the P2X7 pore. To further probe these possibilities, we asked whether CaCC inhibitors were able to inhibit this remaining dye uptake. Co-application of FFA did not inhibit BzATP-induced YO-PRO-1 uptake in rP2X7R-expressing 16F-null cells (Figure 5G), as previously observed in HEK293T cells. Interestingly, 9-AC no longer inhibited dye uptake either (Figure 5G), suggesting that the inhibition of cell permeabilization observed with 9-AC in HEK293T cells (Figure 4C) was related to hTMEM16F. These results suggest that 9-AC inhibits hTMEM16F channels involved in dye uptake and support the hypothesis that remaining dye uptake is likely related to direct passage through the P2X7 pore, although the contribution of another channel cannot be firmly excluded.

To further validate TMEM16F contribution to dye uptake, we asked whether re-expression of TMEM16F along with P2X7 in 16F-null cell lines is able to rescue dye uptake. By assaying varying cDNA ratios between rP2X7 and rat TMEM16F (rTMEM16F) from 1:0.5 to 1:0.001, we found that a ratio of 1:0.05 rP2X7 to rTMEM16F (2 μg rP2X7 and 0.1 μg rTMEM16F) fully rescued dye uptake (Figure 5F). These results therefore not only confirm the contribution of TMEM16F to dye uptake, regardless of the species (human or rat), but also reveal an optimal non-stoichiometric cDNA ratio for restoring dye uptake. Although the underlying mechanism remains elusive, these data are coherent with our findings showing reduced rP2X7 expression in the presence of TMEM16F, which may explain why a cDNA ratio largely skewed in favor of P2X7 is needed to rescue dye uptake.

### 2.8. P2X7 and TMEM16F Are in Physical Proximity

To confirm a physical proximity of P2X7 and TMEM16F, we carried out co-immunoprecipitation in 16F-null cells overexpressing myc tagged rP2X7 and rTMEM16F that was also tagged at its C-terminus with a myc-DDK sequence. Because we found that dye uptake was fully rescued at a cDNA ratio of 1:0.05 in 16F-null cells, and given the fact that P2X7 expression is modulated by the presence of TMEM16F, we asked whether varying this ratio from 1:0.05 to 1:1 may also influence protein expression. We found that co-expressing rP2X7 and rTMEM16F at a 1:1 ratio dramatically reduced total rP2X7 expression, and to a lesser extent TMEM16F expression, in 16F-null cells, but not at a 1:0.05 ratio, as compared to controls (Figure 6A). Thus, in agreement with our previous dye uptake data, these results further confirm the negative effect of TMEM16F on rP2X7 expression (and vice-versa, rP2X7 also seems to decrease TMEM16F expression), and suggest that the ratio 1:0.05 is more suitable for co-immunoprecipitation. In these conditions, we found that anti-FLAG antibody, which recognizes DDK tag of myc-DDK tagged rTMEM16F, specifically and efficiently co-immunoprecipitated rP2X7 subunits, despite rTMEM16F expression being 20 times less than that of rP2X7 (Figure 6B). These data therefore suggest a physical proximity between rTMEM16F and rP2X7.

Next, we asked whether the two proteins co-localize in 16F-null cells by confocal microscopy. We designed a fluorescent rP2X7 that was tagged at its C-terminus with mScarlet, a bright monomeric red fluorescent protein, and transiently co-expressed rP2X7-mScarlet with rTMEM16F at a ratio of 1:0.05 in 16F-nulls cells. By subsequently staining rTMEM16F with a primary anti-h16F antibody coupled to a secondary Alexa 488 conjugated antibody, we showed that both proteins fully co-localized in 16F-null cells (Pearson’s coefficient = 0.897 ± 0.014; *n* = 8 cells from 5 transfections; Figure 6C). Although most of the co-localization appears to be intracellular, putative plasma membrane co-localization was also detected (Figure 6C).

### 2.9. TMEM16F Shapes P2X7 Unitary Conductance

Having revealed the existence of a functional interaction between P2X7 and TMEM16F, we next asked whether P2X7 functions, other than those involved in dye uptake, are also affected in 16F-null cells. We found that current facilitation and current density remained similar to that recorded in HEK293T cells (Figure 7A, left and middle).

A similar variability in the degree of current facilitation was also noticed (Figure 7A, right), as well as a significant increase in the deactivation time constants of facilitated currents (Appendix A). Equally, no change in the profile of whole-cell inhibition by CaCC inhibitors was observed compared to that of HEK293T cells (Figure 7B,C and Appendix A). This result appears surprising given that we previously found 9-AC to no longer inhibit dye uptake in 16F-null cells, and one would expect to see a similar loss of inhibition of whole-cell currents in this cell line. One possibility to explain these data could be that a further 9-AC-sensitive TMEM16 subtype is functionally coupled to P2X7 ion channel function, including current facilitation, but not to dye uptake. Taken together, these data indicate that the absence of hTMEM16F has no apparent impact on whole-cell current properties, and that hTMEM16F has no obvious contribution to current facilitation.

Single-channel recordings revealed, however, the disappearance of the O2 conductance state observed in HEK293T cells, while the O1 conductance state remained present in seven out of nine patches (Figure 7D,E). In the remaining two patches, an additional unitary current of greater magnitude than that corresponding to the O2 conductance state was recorded. However, as this current occurred only very occasionally, we cannot attribute it definitively (Figure 7D,E). As a result of these changes in unitary currents, the mean single-channel conductance was significantly lower in 16F-null cells than in HEK293T cells (Figure 7F). This decrease in mean single-channel conductance may compensate for the increased cell surface expression of P2X7 subunit in 16F-null cell lines (Figure 5C), thus accounting for the apparently unchanged whole-cell current density. In addition, these unitary currents were strongly inhibited by TA and AZ10606120, but less so by FFA and 9-AC (Figure 7G). Collectively, our data suggest that endogenous hTMEM16F shapes the O2 single-channel conductance recorded in rP2X7-expressing HEK293T cells.

## 3. Discussion

In this work, we shed new light on the molecular mechanisms underlying macropore formation and agonist-evoked current facilitation, two hallmarks of P2X7 that have remained enigmatic for several decades. We reveal that both phenomena implicate functional complexes formed between P2X7 and TMEM16 channels (Figure 8). We also provide evidence that current facilitation does not stem from pore dilation. Indeed, our single-channel data reveal an increase of P2X7 open probability following current facilitation, but not the progressive appearance of a new conducting state.

In agreement with previous work [30,31], we confirm that P2X7 and TMEM16 channels are able to form functional complexes in both HEK293T and *Xenopus* oocytes. In HEK293T cells, we identify the subtype TMEM16F as part of these functional complexes involved in dye uptake and show that the TMEM16F and P2X7 channels are located in proximity to one another. However, whether they directly interact through protein-protein contacts, or indirectly through Ca^2+^ signaling requires further experimental testing. In addition, because all experiments described in this study were carried out in overexpressing systems, overexpressed P2X7 and TMEM16F may be misplaced in terms of cellular localization and show altered behavior that may not be exactly representative of the native interaction. Therefore, further work is also needed to demonstrate that this functional interaction is also relevant to endogenous proteins.

The first important finding of this study is that TMEM16F has important implications for both P2X7 expression and function. We show that TMEM16F influences P2X7 expression, suggesting that TMEM16F may act as a regulating element of P2X7 expression. Although the underlying mechanism is currently unknown and deserves additional studies, one possibility could be that TMEM16F regulates P2X7 transcriptional activity. This hypothesis is consistent with the fact that TMEM16F strongly reduces both total and cell-surface expression of P2X7. We also show that native TMEM16F contributes to the function of P2X7, through the occurrence of an O2 conducting state, that is not observed in the absence of TMEM16F. The significance of this state is currently unknown. One may argue that currents originating from the O2 state stem from TMEM16F channel, but this possibility seems unlikely because the activity of TMEM16F, which is an outwardly-rectifying channel, is expected to be minimal at negative potentials, as used in our protocols to record P2X7 activity [44,45]. Another possibility could be that the close physical proximity of TMEM16F to P2X7 somehow influences single-channel conductance characteristics.

The lack of action of 9-AC on BzATP-induced P2X7 currents in CaCC-void Axolotl oocytes provides strong evidence that this inhibitor does not directly bind to P2X7. However, the fact that it inhibits P2X7 currents in both HEK293T cells and *Xenopus* oocytes suggests instead that 9-AC binds to endogenous TMEM16 channels and inhibits P2X7 activity via an allosteric interaction, due to a close juxtaposition of the two channels (Figure 8). Our data further suggest that 9-AC inhibits TMEM16F channels involved in dye uptake. Yet, the fact that 9-AC no longer inhibits dye uptake in 16F-null cells, while still inhibiting BzATP-evoked currents, suggests that another TMEM16 subtype is coupled to P2X7 ion channel function, including current facilitation, but not to dye uptake (Figure 8). 9-AC is a non-selective inhibitor, and therefore additional experiments with more specific TMEM16 inhibitors are needed to identity this (these) subtype(s).

Our data also suggest that a significant fraction of P2X7 resides outside of functional complexes made with TMEM16F. We show that 9-AC does not abolish BzATP-evoked currents, even after a 1-min long application (Figure 3A), thus raising the possibility that a fraction of “free” P2X7 channels (i.e., those not embedded in functional complexes) reside at the cell membrane (Figure 8).

Conversely, the mode of action of FFA and TA remains unclear. Although FFA inhibition was observed in *Xenopus* oocytes with no direct action on P2X7, its inhibitory effect was variable in HEK293T cells, and no definitive conclusion can be drawn. For TA, however, P2X7 currents were systematically abolished following application of the inhibitor, even in Axolotl oocytes, suggesting that it may either directly inhibit P2X7 channel or act non-specifically through a currently unknown mechanism. Regardless of the mechanism at play, this result is in apparent disagreement with previous work having found that co-application of TA and ATP to human microglia did not inhibit endogenous P2X7 channel activity [13]. The reasons behind this discrepancy are unknown but may result from factors differing between recombinant and native systems. Finally, our data demonstrate that TA is not suitable for dye uptake experiments because of its fluorescence quenching properties, a finding that is also supported by a very recent study [46].

The second important finding of the present work is that the functional interplay between P2X7 and members of the TMEM16 family largely contributes to current facilitation, with no obvious contribution of the TMEM16F subtype. This is supported by the fact that 9-AC strongly impairs BzATP-evoked current facilitation in both HEK293T and 16F-null cells. Interestingly, TMEM16 channels were also previously suggested to contribute to current facilitation in *Xenopus* oocytes through a different mechanism [31]. The authors suggested that current facilitation results in the secondary activation of TMEM16A Cl^−^ channels that follows P2X7 activation and subsequent Ca^2+^ flux. However, we show that in HEK293T cells the mechanism underlying current facilitation results in a dramatic increase of P2X7 N*P*o, with no evidence of the appearance of a new, secondary unitary conductance that would correspond to the activation of another channel. Therefore, it appears that different mechanisms underlying current facilitation are at play depending on the cell type. The two mechanisms can co-exist, but at present we do not know whether those TMEM16 Cl^−^ channels are active in our conditions.

In close agreement with previous data [7,8], our results suggest that plasma membrane cholesterol is a negative allosteric modulator of P2X7. It maintains the functional complex-embedded P2X7, and possibly “free” P2X7, in a low channel activity state in response to initial, brief agonist application. Longer or repeated agonist application then switches channels to higher activity states, which, in turn, underlies the observed increased current amplitudes and slowed deactivation kinetics. As suggested previously [6,8], we too propose that this effect is likely attributable to a progressive dissociation of cholesterol from channels (Figure 8). Molecular mechanisms underlying agonist-evoked cholesterol dissociation remain poorly understood, but one possibility would be an activity-dependent increase of membrane mobility of complexes from high (e.g., lipid raft) to low cholesterol content [47]. Whether cholesterol binds directly to P2X7, TMEM16 channels or the functional complex as a whole, remains to be determined. A recent study has demonstrated that cholesterol inhibits P2X7, most likely through interactions with membrane palmitoyl moieties attached to cysteine residues located in an intracellular juxtamembrane cysteine-rich domain of P2X7 [8]. Supported by P2X7 cryo-EM structures [29], one possibility is that cholesterol binding to those competing transmembrane sites interferes with P2X7 channel gating, most likely by stabilizing the closed channel state, and/or by destabilizing the open channel state.

The third important finding of this study is that large molecule permeability proceeds through at least two components in HEK293T cells (Figure 8). The first one (component 1 in Figure 8) likely entails a direct passage through the P2X7 pore, as suggested previously [8]. Cryo-EM structures of P2X7 suggest the existence of a finite ~5 Å open channel diameter, with no evidence of larger sized pores [29]. The minimal cross-section of YO-PRO-1 being ~7Å, which exceeds the 5 Å-outer limit of the P2X7 open pore, suggests that direct passage of YO-PRO-1 would occur at extremely low rates, as recently suggested in the case of P2X2, for which permeation rates would be largely below to 10^6^ molecules per second per channel [26]. We therefore suggest that molecules smaller than YO-PRO-1, such as NMDG^+^, robustly diffuse through the P2X7 open pore, as revealed previously [26], albeit at slower permeation rates than those of Na^+^. However, for larger molecules, such as YOYO-1 (763 Da) or TO-TO-1 (894 Da) for which P2X7-dependent cell permeabilization has been described [11,14], direct permeation through the resolved P2X7 open channel structure seems challenging [29]. Therefore, the contribution of a second, non-exclusive component (component 2 in Figure 8) that involves TMEM16F channels (and perhaps other TMEM16 members) represents an alternative permeation mechanism of larger nanometer-sized molecules. Importantly, as this second component is linked to a weakly expressed TMEM16F (relative to P2X7), our data may explain why knockdown of TMEM16F by siRNAs, for which low expression levels can escape silencing, does not apparently affect P2X7 dye uptake in macrophages [14] and HEK cells [30]. Our data thus provide a possible explanation for previous data that have remained otherwise difficult to explain [9,10,14,16]. In light of our results showing that long agonist exposure increases P2X7 N*P*o, we speculate that components 1 and 2 are boosted during long agonist exposure, and thus both contribute to the enhanced levels of large molecule cell membrane permeability observed in dye uptake experiments.

The precise permeation pathway underlying large molecule permeation of component 2 is currently unknown, but may involve the TMEM16F pore, due to its function as a poorly selective ion channel [44,48], enabling the passage of small organic cations such as NMDG^+^ [44]. Another possibility is that dye uptake shares the same mechanisms as those implicated in TMEM16F scramblase activity [44,45,49]. Whatever the mechanism, the identification of the dye uptake pathway of component 2 merits further study.

In conclusion, our data reveal that P2X7 can form functional complexes with TMEM16 channels in HEK293T cells. Interestingly, a very recent work has also reported similar modulation of P2X7 activity with another channel, TMEM163 [50], suggesting that P2X7 may functionally associate with different kinds of ion channel families. We propose that the functional complexes formed between P2X7 and TMEM16 channels act as a regulating hub, which, upon P2X7 activation and dependent on the surrounding membrane cholesterol level, orchestrates a hive of activity, eliciting not only channel gating (e.g., efflux of K^+^ and Ca^2+^ signaling), but also other cell-specific signaling, including membrane blebbing, interleukin release and phospholipid scrambling [9]. Given the critical roles of P2X7 and TMEM16 channels in many diseases, this platform may be involved in important pathophysiological processes that lead, for example, to inflammation [51] and mechanical allodynia [52,53], and may therefore be therapeutically important. Future work is needed to unravel its broader implications in disease.

## 4. Materials and Methods

### 4.1. Cell Culture and Transfection

HEK293T cells (ATCC) were cultured in Dulbecco’s modified Eagle’s medium, supplemented with 10% fetal bovine serum, 1× GlutaMax, 100 units/mL penicillin and 100 μg/mL streptomycin (Gibco Life Technologies). For HEK293T hTMEM16F-null cells, this medium was further supplemented with 1 μg/mL puromycin dihydrochloride (Gibco Life Technologies).

Trypsin-treated cells were seeded onto 9–12 mm glass coverslips (VWR) in 35 mm dishes for patch-clamp and fluorescence experiments, or in 100 mm dishes for biochemical experiments, in both cases pre-treated with poly-l-lysine (Sigma Aldrich, St. Louis, MO, USA). Cells were incubated at 37 °C and in presence of 5% CO_2_.

Transfections were carried out using the calcium phosphate precipitation method. The cDNA encoding rat *P2rx7* gene (ID Q64663) constructs and enhanced Green Fluorescent Protein construct (eGFP) were contained within pcDNA3.1(+) plasmids (Invitrogen). rP2X7-mScarlet was designed by tagging mScarlet (from gene synthesis) in the C-terminus of rP2X7 within a pcDNA3.1(+) vector. rTMEM16F cDNA was contained within a pCMV6 vector featuring a myc-DDK tag in the C-terminus (OriGene, RR212483). This myc-DDK tag containing the sequence DYKDDDDK can be recognized by anti-FLAG antibody. For whole-cell patch clamp experiments, cells were co-transfected with the rP2X7 construct (0.5–0.8 μg) and an eGFP (0.3 μg) which allowed to identify cells having undergone efficient transfection. For single-channel experiments, the quantity of transfected rP2X7 construct DNA was reduced to 0.01–0.08 μg. Cells were washed one day after transfection with PBS solution, and the medium replaced with fresh. For biochemical experiments, each 100 mm dish was either co-transfected with cDNAs encoding myc tagged rP2X7 (5 μg) and myc-DDK tagged rTMEM16F constructs (5 or 0.25 μg, as indicated) or transfected with each construct alone (10 μg). For dye uptake rescue experiments, cells were transfected with 2 μg myc tagged rP2X7 construct either alone or co-transfected with 1, 0.1, 0.01 or 0.002 μg myc tagged rTMEM16F. For co-localization by confocal microscopy, cells were transfected with 2 μg myc tagged rP2X7 and 0.1 μg myc tagged rTMEM16F.

### 4.2. Oocytes Preparation and Injection

Ovarian lobes were surgically removed from *Xenopus laevis* and *Ambystoma mexicanum* Axolotl females and oocytes were isolated as described [54,55]. After nuclear injection of 300 pg of rat P2X7 construct, oocytes were incubated in Barth’s solution containing 1.8 mM CaCl_2_ and gentamycin (10 mg/mL) at 19 °C for 1–3 days before electrophysiological recordings.

### 4.3. Ethics Approval

All experimental procedures complied with official European guidelines for the care and use of laboratory animals (Directive 2010/63/UE) and were approved by the ethical committee of Bordeaux and French MESRI ministry. All animals were treated humanely and with regard for alleviation of suffering.

### 4.4. Molecular Biology

CRISPR/Cas9 method was carried out as previously described [56,57]. The oligonucleotide encoding the sgRNA sequence (5′-AATAGTACTCACAAACTCCG-3′), which targets exon 2 of *TMEM16F* gene, containing BbsI overhangs was cloned into BbsI sites in pSpCas9(BB)-2A-Puro (Addgene plasmid ID:48139). The plasmid obtained was transfected into HEK293T using the calcium phosphate precipitation method. After 48 h of transfection, 1 μg/mL puromycin was applied to select cells for 72–96 h, with medium change every 24 h. Then, transfected cells were serial-diluted in 96-well plates to select for single-cell colonies. After 14–21 days, the single-cell colonies were expanded and screened for the absence of hTMEM16F. Verification was carried out by sequencing (Eurofins Genomics, see Appendix A) and Western blot with the rabbit anti-hTMEM16F antibody (Sigma Aldrich, HPA038958) (see Figure 5A).

### 4.5. Patch-Clamp Electrophysiology

Whole-cell recordings were carried out 24–48 h after transfection, and single-channel recordings 24 h after transfection. Patch pipettes were pulled from borosilicate glass capillaries (Harvard Apparatus, Holliston, MA, USA) to yield resistances of 3–5 MΩ for whole-cell recordings, and 15–20 MΩ for single-channel recordings. For single-channel recordings, pipettes were coated with Sylgard 184 (Dow Corning Co., Midland, MI, USA) and fire polished before use. pH was verified for all solutions used, and if necessary, adjusted carefully using NaOH to pH 7.32–7.33. All solutions were maintained at approximately 300 mOsm.

Cells were voltage-clamped to −60 mV (whole-cell) or −120 mV (outside-out) using the EPC10 amplifier (HEKA, Reutlingen, Germany), and data were recorded with PATCHMASTER software. Data were acquired at 10 kHz and low-pass filtered at 2.9 kHz. Applications of agonist and/or inhibitor were carried out by perfusion, using three capillary tubes placed directly over the cell/patch of interest. These capillaries are displaced using the SF-77B Perfusion Fast Step system (Warner), ensuring solution exchange within 5–10 ms. For whole-cell recordings, Normal Extracellular Solution (NES) contained 140 mM NaCl, 2.8 mM KCl, 1 mM CaCl_2_, 0.1 mM MgCl_2_, 10 mM HEPES, 10 mM Glucose, pH 7.32–7.33. Patch pipettes contained 140 mM KCl, 5 mM MgCl_2_, 5 mM EGTA, 10 mM HEPES, adjusted to pH 7.3 with NaOH. For single-channel recordings, NES contained 147 mM NaCl, 2 mM KCl, 1 mM CaCl_2_, 0.1 mM MgCl_2_, 10 mM HEPES, 10 mM Glucose adjusted to pH 7.3 with NaOH. Intracellular solution contained 147 mM NaF, 10 mM HEPES and 10 mM EGTA, adjusted to pH 7.3.

2′(3′)-*O*-(4-benzoylbenzoyl) adenosine 5′-triphosphate (BzATP) (triethylammonium salt, Sigma Aldrich) was used as P2X7 agonist at a concentration of 10 μM. TMEM16 inhibitors used were the following: tannic acid (TA) 20 μM, flufenamic acid (FFA) 100 μM, 9-anthracene-carboxylic acid (9-AC) 1 mM. Specific P2X7 antagonist AZ10606120 (dihydrochloride salt, Tocris) was used at a concentration of 1 μM. BzATP and TA were solubilized in NES. In the case of AZ10606120, FFA and 9-AC, concentrated stock solutions were firstly produced in DMSO, which were subsequently diluted in NES to achieve the desired working concentration in ≤0.1% DMSO. For control application (without inhibitors), 0.1% DMSO was added in NES. Owing to the facilitation effect produced by prolonged agonist exposure, only one cell was patched per coverslip, to ensure that receptors studied are indeed “naïve” upon the first agonist stimulation. For cells and patches treated by methyl-β-cyclodextrin (MCD), coverslips were incubated with 50 μL of 15 mM MCD (Sigma Aldrich) solubilized in DMEM for 15 min at 37 °C. This volume ensures complete coverage of the coverslip with MCD solution. Following incubation, MCD solution is removed by gentle washing with NES, before being used for electrophysiological experiments. In facilitated conditions, perfusion of BzATP (10 μM) was carried out for 30 s while in the cell-attached configuration, before piercing of the membrane and excising of the patch to the outside-out configuration, in which recordings were carried out.

### 4.6. Xenopus and Axolotl Oocyte Electrophysiology

TEVC recordings were performed as previously described [54,55]. Briefly, recordings were carried out at room temperature using glass pipettes (1–2 MΩ) filled with 3 M KCl solution to ensure reliable holding potentials. Oocytes were voltage-clamped at −60 mV and membrane currents were recorded with an OC-725B amplifier (Warner Instruments) and digitized at 1 kHz on a Power PC Macintosh G4 using Axograph X. Oocytes were perfused at a flow rate of 10–12 mL/min with Ringer solution, pH 7.4 containing in mM: 115 NaCl, 3 NaOH, 2 KCl, 1 CaCl_2_, and 10 HEPES. BzATP and TMEM16 inhibitors were solubilized in Ringer solution and used at the same concentration as in patch-clamp experiments. Agonist and drugs were applied using a computer-driven valve system (Ala Scientific, Farmingdale, NY, USA).

### 4.7. Co-Immunoprecipitation, Cell Surface Biotinylation Assays and Western Blotting

For co-immunoprecipitation, non-transfected or transfected 16F-null cells with myc tagged rP2X7 and myc-DDK tagged rTMEM16F at a ratio of 1:0.05, were lysed by incubation with gentle agitation for 80 min at 4 °C with lysis buffer containing 20 mM HEPES, 100 mM NaCl, 1% Triton-X, 5 mM EDTA, Pierce Protease Inhibitor Tablets (Thermo Fischer, Waltham, MA, USA). After 10 min of centrifugation at 14,000 rpm the supernatant was collected. Input samples were collected at this stage. After an initial pre-clearing process of cell lysate with Protein G Sepharose Fast Flow resin (Sigma Aldrich, see manufacturer’s instructions), 5 μg anti-FLAG mouse antibody (Sigma Aldrich, F1804) was added to the cell lysate and left under gentle agitation for 2 h at 4 °C. Protein G Sepharose Fast Flow resin was then added and left under gentle agitation for 1 h at 4 °C. Following this, the resin was washed three times with washing buffer containing 20 mM HEPES, 500 mM NaCl, 1% Triton-X, 5 mM EDTA, Pierce Protease Inhibitor Tablets (Thermo Fischer), and two times with lysis buffer. The resin was then resuspended in NuPage LDS Sample Buffer x1 (Thermo Fischer) and 70 mM DTT, and boiled for 10 min. The supernatant was loaded onto NuPage Novex Bis-Tris 4%–12% gel (Thermo Fischer) and migrated in MOPS buffer. Transfer onto nitrocellulose membrane was carried out using the TransBlot Turbo system (BioRad) and the membrane blocked for 30 min in TPBS (PBS supplemented by 1% dried non-fat mik, 0.5% BSA, 0.05% Tween-20). The membrane was incubated overnight at 4 °C with TPBS containing either anti-c-myc mouse antibody, dilution 1:500 (Invitrogen, Thermo Fischer, reference 13–2500) or anti-β-actin mouse antibody, dilution 1:5000 (Sigma Aldrich, reference A5441). Three washes with TPBS were carried out, before a second incubation in TPBS containing HRP-conjugated sheep anti-mouse antibody, dilution 1:10,000 (GE Life Sciences, reference NA9310) for 1–2 h at room temperature. Three further washes with TPBS were carried out before revelation using Amersham ECL Select Western Blotting Detection Reagent (GE Life Sciences). Chemiluminescence was measured using the Amersham Imager 600. Images have been cropped for presentation; original uncut images are available in Appendix A.

For cell surface biotinylation, transfected HEK293T and 16F-null cells were washed three times with PBS+ (PBS supplemented with 1 mM MgCl_2_ and 0.4 mM CaCl_2_, adjusted to pH 8.0), and incubated 30 min with 2 mM sNHS-SS-Biotin (Thermo Fisher, reference 21,331) in PBS+. Then, a washout step with PBS+ was carried out, before quenching any excess sNHS-SS-Biotin for 10 min with 20 mM Tris in PBS+. Three further washing steps with PBS were carried out before lysis step. Following lysis (as described above), neutravidin-agarose resin (Thermo Fisher, reference 29,200) was added to the cell lysate and incubated under gentle agitation overnight at 4 °C. Samples were then prepared as described for co-immunoprecipitation and loaded onto Mini-PROTEAN TGX 4–15% and migrated in TGS buffer. Western blotting was carried out as described above.

### 4.8. Immunocytochemistry

Transfected or non-transfected 16F-null cells were fixed 24 h post transfection in 4% PFA (PBS, 4% paraformaldehyde, pH 8.0) for 15 min at room temperature. Coverslips were then washed in PBS for 5 min and permeabilized for 30 min in block solution (PBS, 1% BSA, 0.1% Triton X-100). Coverslips were incubated at 4 °C overnight with primary anti-hTMEM16F rabbit antibody (Sigma Aldrich, HPA038958) diluted 1:500 in block solution. Coverslips were then washed in PBS and incubated with secondary fluorescent probe-conjugated antibody (goat anti-rabbit IgG H&L (Alexa Fluor^®^ 488) (ab150077, Abcam, Cambridge, UK) diluted 1:500, and Hoechst stain (4 μg/μL) for 1 h at RT in block solution. Coverslips were then mounted using mounting reagent (Prolong Gold antifade, P36930, Invitrogen) and allowed to dry overnight prior to imaging with Leica SPE.

### 4.9. Fluorescence Microscopy

Video fluorescence microscopy measurements were carried out using a Leica FW4000 and 40× objective (Platform of Quantitative Imagery, Faculté de Pharmacie, Université de Strasbourg). Acquisition of fluorescence images was carried out at an interval of every 5 s using the software MetaMorph (Molecular Devices). The experiment was divided into two acquisition periods; an initial acquisition of 10 min where cells were incubated in NES/YO-PRO-1 solution where YO-PRO-1 is at a concentration of 10 μM (iodide salt, Thermo Fischer). The solution was then gently exchanged for a NES/YO-PRO-1 solution containing BzATP and inhibitors (or BzATP alone) for a second acquisition period of 15 min. This second solution containing inhibitors was prepared immediately prior to application, and concentrations used are the same as those used for electrophysiology. Cells were maintained at 37 °C during measurements.

Confocal imaging was captured with Leica SPE using oil immersion objective: 63×, Numerical Aperture 1.4. Excitation (λexc) and emission (λem) wavelengths were as follows: Hoechst (λexc = 364 nm; λem = 430–481 nm), Alexa 488 (λexc = 495 nm; λem = 500–600 nm) and mScarlet (λexc = 561 nm; λem = 648–708 nm).

### 4.10. Fluorescence Spectroscopy

Fluorescence spectra were obtained on SAFAS Xenius Spectrophotometer. Measurements were made in quartz cuvettes. A 1 mL solution was excited at 480 nm. Solutions contained NES supplemented with 10 μM YO-PRO-1 alone (−dsDNA in Figure 4D), or 10 μM YO-PRO-1 complexed to 29 μg dsDNA (+dsDNA), or 10 μM YO-PRO-1 complexed to 29 μg dsDNA with inhibitors (+dsDNA +inhibitors). Inhibitor concentrations used were the same as those used for electrophysiology. The bandwidth for excitation and emission was 10 nm.

### 4.11. Data Analysis

For electrophysiological data analysis, FitMaster (HEKA Electronics v2 × 73 × 1) and Igor Pro (WaveMetrics, v6.32A) were used. Experiments were repeated several times, over at least two independent transfections, with precise details given in figure legends. The extent of facilitation (as in Figure 7A) was determined by calculating the fold difference between the 8th agonist-evoked current and initial agonist-evoked current in a series of 8 repeated applications. The current density (as in Figure 7A) was calculated as the pA/pF of the 8th agonist-evoked current in a series of 8 repeated applications.

For single channel data analysis, TAC and TACFit (Bruxton) were used. Data were re-filtered offline to give a final f_c_ of 100 Hz. Dead time (t_d_) was set to 1.8 ms and transitions shorter than t_d_ were ignored. Transitions longer than t_d_ were accepted as events. Single-channel current amplitudes were determined by all points histograms fitted to Gaussian distributions, using maximum likelihood methods:(1)$$f\left(I\right)=\sum_{i=1}^{n}\frac{a_{i}}{\sigma_{i}\sqrt{2\pi }}\exp\left[-\frac{{\left(I-A_{i}\right)}^{2}}{2\sigma_{i}^{2}}\right],$$
where *f*(*I*) is the total probability density of a given amplitude value *I*, *A_i_* is the *i*th channel amplitude, σ*_i_* is the standard deviation of the *i*th channel amplitude, and *a_i_* is the fraction of the data represented by the *i*th amplitude. Conductance was determined by dividing current amplitudes by the holding potential (−120 mV).

Only patches featuring three or fewer channels were analyzed, and analysis of stacked events resulting from simultaneous channel openings was less than 5% of the total events analyzed. The number of channels present within a given patch can be determined by visual inspection of the maximum number of coinciding stacking events. These stacked openings are designated as such, in order to verify the proportion that they represent of the total events analyzed. Assuming N independent and identical channels in the patch, each with a probability *P*o of being open, the probability that *k* channels open simultaneously is given by the binomial distribution:(2)$$P\left(k\right)=\left(\begin{array}{c} N\\ k \end{array}\right)P_{o}^{k}{\left(1-P_{o}\right)}^{N-k}.$$

If one assumes that the O2 state results in the simultaneous opening of two O1 states (*k* = 2), then its probability of occurring is equal to 0.0003, taking N = 3 (we selected only patches containing three or fewer channels for analysis) and *P*o = 0.01 (0.03/3 = 0.01 with N*P*o experimentally determined as 0.03 for O1 in control conditions). This theoretical value is largely below the actual value of 0.05 we determined for the O2 state (that is 0.14/3 ≈ 0.05 with N*P*o = 0.14 for O2 state in control conditions), suggesting that O2 did not originate from stackings of double O1 openings.

For control conditions, we recorded from 7 different outside-out patches, in which 1518 events were analyzed. Each patch contained between 1 to 15 sweeps. For facilitated conditions, we recorded from 5 outside-out patches, in which 754 events were analyzed (ranged between 3 to 19 sweeps). For MCD-treated cells, we recorded single-channel currents from 7 outside-out patches excised from cells that were pre-treated with MCD, and in which 396 events were analyzed (ranged between 1 to 13 sweeps).

Detection of events was carried out by using 50% of the single-channel current amplitude as the detection threshold. Dwell time histograms were fitted by the minimum number of exponential functions according to the following equation:(3)$$f\left(t\right)=\sum_{i=1}^{n}a_{i}/\tau_{i}\exp\left(-t/\tau_{i}\right),$$
where *t* is the time, *τ_i_* is the time constant, and *a_i_* is the relative area (the sum of all areas is equal to (1). The mean open time for a selected state is given by:(4)$$\tau_{state}=\sum a_{i}\tau_{i}$$
(5)$$f\left(t\right)=\sum_{i=1}^{n}a_{i}/\tau_{i}\exp\left(-t/\tau_{i}\right).$$

For conductance analysis, the same patches were analyzed as in mean open time analysis, with the addition of supplementary patches. These supplementary patches contained too many stacked events to be considered for mean open time analysis but remain suitable for unitary conductance analysis.

For analysis of percentage inhibition of rP2X7 single-channel currents by CaCC inhibitors, a weighting is established of open-channel activity compared to closed-channel activity (within the 10 s period of drug application, illustrated in Figure 3C by grey shading). This weighting is calculated for recordings in patches exposed solely to BzATP, and then for subsequent recordings of the same patch exposed to BzATP and CaCC inhibitor co-applications. In the case of patches which did not exhibit instant inhibition, BzATP and CaCC inhibitor co-applications were repeated until a steady-state level of inhibition was achieved. These steady-state inhibitions were then averaged. A percentage inhibition is therefore calculated for each patch, which are subsequently averaged.

Deactivation currents (decay currents when BzATP is removed) were fit by single exponentials according to the following equation:(6)$$f\left(t\right)=A\;\exp\left(-t/\tau_{d}\right),$$
where *t* is the time, *τ_d_* is time constant of deactivation, and A is the current amplitude.

For fluorescence experiment analysis, Fiji and Igor Pro were used. For co-localization analysis Pearson’s correlation coefficient was measured using Coloc2 in ImageJ.

### 4.12. Statistical Analysis

Data are reported as mean values ± SEM. All experiments were replicated over a minimum of 2 transfections and 3 cells. For electrophysiological data, exact data points are shown as dots within histograms, and for fluorescence data, the number of cells is shown in brackets. For statistical analysis, GraphPad Prism software (v8.0.2) was used. The normality of data distribution was tested before selection of an appropriate test of statistical significance and *p* < 0.05 was considered as significant difference. For electrophysiological and Western blot data, Mann-Whitney or two-tailed (paired or unpaired, when indicated) Student’s *t*-test was used. For fluorescence measurements, one-way Kruskal-Wallis analysis of variance test followed by Dunn’s multiple comparison post-hoc test or Mann-Whitney test was employed, as indicated in figure legends.

### 4.13. Data Availability

All data and associated protocols for this study are available in the main text and Supporting information. Materials may be requested to T.G.

## Figures and Tables

**Figure 1 ijms-22-06542-f001:**
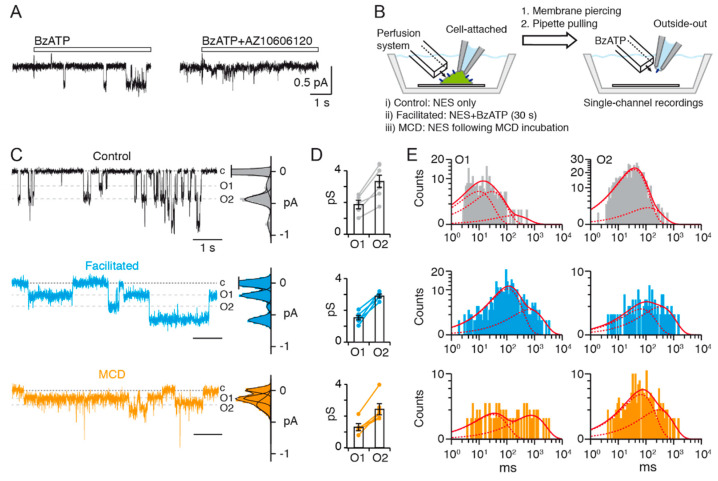
Current facilitation of P2X7 results in an increase of single-channel open probability. (**A**) Single-channel currents elicited by 10 μM BzATP (duration of application indicated by the white bar) from an outside-out patch of HEK293T cell transiently transfected with rP2X7 (left) are inhibited by 1 μM AZ10606120 when co-applied with 10 μM BzATP (right). Traces are from the same patch and channel openings are downward deflections. (**B**) Cartoon depicting the protocol employed for establishing excised patches from cells expressing rP2X7 (blue ovals) in: (i) control conditions, (ii) facilitated conditions (10 μM BzATP for 30 s), and (iii) MCD-treated conditions. NES: normal extracellular solution. Figure not to scale. (**C**) Single-channel currents elicited by 10 μM BzATP from outside-out patches of HEK293T cells transiently transfected with rP2X7. Cells were either not treated (black), or pre-treated with a 30 s perfusion of 10 μM BzATP (blue), or incubated with MCD (orange) prior to membrane excising. Dashed grey lines indicate unitary currents corresponding to conductance states O1 and O2. Dotted lines indicate basal current corresponding to closed channels, labeled c. Corresponding all points histograms of currents are shown right of the traces. Distributions were fit by a sum of Gaussians. (**D**) Summary of corresponding unitary conductances. Linking lines indicate data points that originate from the same patches (*n* = 7 patches for control, 6 for facilitated and 7 for MCD-treated conditions). Bars represent mean ± SEM. (**E**) Corresponding open dwell-time histograms fitted by the sum (solid red line) of several exponential functions (dashed red lines). For control patches (grey histograms), τ_1_ = 9.6 ms, relative area, a_1_ = 0.49; τ_2_ = 29 ms, a_2_ = 0.48; τ_3_ = 180 ms, a_3_ = 0.03; mean open time, τ_O1_ = 24 ms for O1, and τ_1_ = 38 ms, a_1_ = 0.92; τ_2_ = 119 ms, a_2_ = 0.08; τ_O2_ = 44 ms for O2. For facilitated (blue histograms), τ_1_ = 108 ms, a_1_ = 0.77; τ_2_ = 654 ms, a_2_ = 0.23; τ_O1_ = 234 ms for O1 and τ_1_ = 68 ms, a_1_ = 0.46; τ_2_ = 332 ms, a_2_ = 0.54; τ_O2_ = 211 ms for O2. For MCD-treated patches (orange histograms), τ_1_ = 32 ms, a_1_ = 0.52; τ_2_ = 752 ms, a_2_ = 0.48; τ_O1_ = 378 ms, for O1 and τ_1_ = 66 ms, a_1_ = 0.70; τ_2_ = 294 ms, a_2_ = 0.30; τ_O2_ = 134 ms for O2. All data were recorded at −120 mV and filtered at a final f_c_ of 100 Hz.

**Figure 2 ijms-22-06542-f002:**
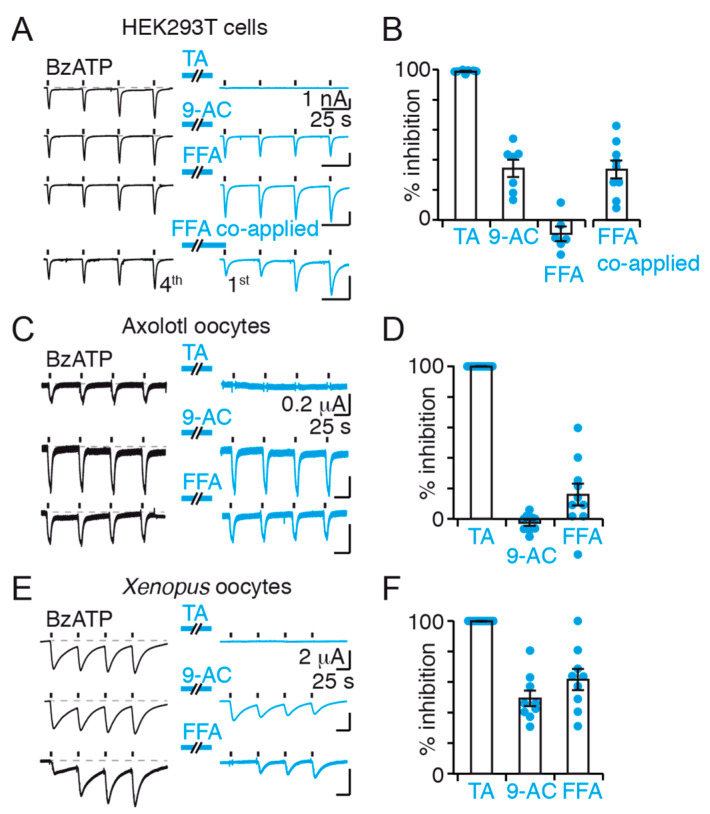
Functional effects of CaCC inhibitors on P2X7 currents in HEK293T cells, Axolotl and *Xenopus* oocytes. (**A**) Whole-cell currents evoked by repeated 2 s applications of 10 μM BzATP (black bars) before and after 60 s perfusion of TMEM16 inhibitors (blue bars) in HEK293T cells transiently expressing rP2X7. In the case of FFA co-applied, perfusion was followed by a co-application with 10 μM BzATP. (**B**) Summary of whole-cell inhibition (*n* = 7 cells for TA, 7 for 9-AC, 6 for FFA, and 9 for FFA co-applied). Inhibition is calculated by comparing current amplitude of the fourth application before and first application after perfusion. Bars represent mean ± SEM. (**C**) Whole-cell BzATP-evoked currents recorded by TEVC of Axolotl oocytes expressing rP2X7 before and after application of TMEM16 inhibitors using the same protocol as in panel A. (**D**) Summary of inhibition in Axolotl oocytes calculated as in panel B (*n* = 10 oocytes for TA, 9 for 9-AC, and 10 for FFA). Bars represent mean ± SEM. (**E**) Whole-cell BzATP-evoked currents recorded by TEVC of *Xenopus* oocytes expressing rP2X7 using the same protocol as in panel A. (**F**) Summary of inhibition in *Xenopus* oocytes calculated as in panel B (*n* = 10 oocytes for TA, 9 for 9-AC, and 9 for FFA). Bars represent mean ± SEM. CaCC inhibitor concentrations were 20 μM TA, 1 mM 9-AC and 100 μM FFA. Data were recorded at −60 mV.

**Figure 3 ijms-22-06542-f003:**
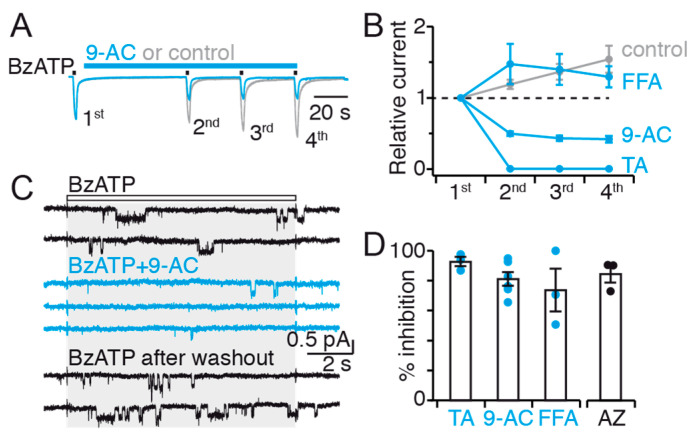
TA and 9-AC prevent current facilitation and inhibit single-channel activity. (**A**) Whole-cell currents evoked by repeated 2 s applications of 10 μM BzATP in the presence (blue) or absence of 9-AC (grey) in HEK293T cells transiently expressing rP2X7. Currents are normalized to the first application. (**B**) Summary of currents relative to the first application in the absence (control) or presence of inhibitors (*n* = 4 cells for TA, 10 for 9-AC, 8 for FFA, and 8 for control). Symbols represent mean ± SEM. (**C**) Single-channel rP2X7 currents elicited by 10 μM BzATP from an outside-out patch of HEK293T cell (black traces) are inhibited by 1 mM 9-AC when co-applied with 10 μM BzATP (blue traces). Shown sweeps were sequentially recorded from the same patch. Note that inhibition is reversible after washout. Duration of application is indicated by the white bar and grey shading. Data were filtered at a final f_c_ of 100 Hz. (**D**) Summary of single-channel inhibition (*n* = 3 patches for TA, 6 for 9-AC, 3 for FFA, and 3 for AZ10606120). Bars represent mean ± SEM. See Materials and Methods for calculation. Inhibitor concentrations were 20 μM TA, 1 mM 9-AC, 100 μM FFA and 1 μM AZ10606120. Single-channel and whole-cell data were recorded at −120 and −60 mV, respectively.

**Figure 4 ijms-22-06542-f004:**
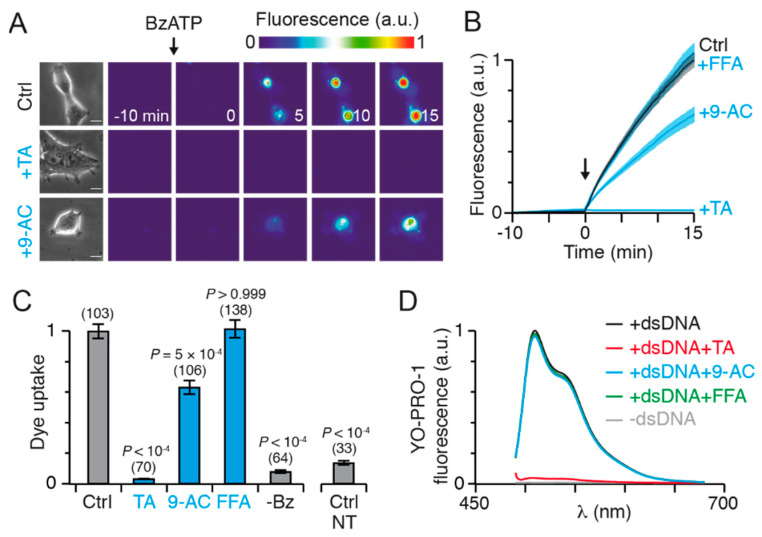
Effects of CaCC inhibitors on dye uptake. (**A**) Time series fluorescence images of YO-PRO-1 dye uptake in HEK293T cells transiently expressing rP2X7, in the absence or presence of inhibitors, as indicated. Fluorescence (in arbitrary units) was acquired at indicated times before and after 10 μM BzATP (arrow) application, which begins at 0 min. Inhibitors were co-applied at the same time as BzATP. On the left are shown the corresponding microphotographs under transmitted light (scale bar, 10 μm). (**B**) Corresponding YO-PRO-1 dye uptake over time upon BzATP activation (arrow) in the absence or presence of inhibitors, as indicated. Solid lines represent mean ± SEM (shaded areas). (**C**) Summary of BzATP-induced YO-PRO-1 dye uptake normalized to rP2X7-transfected cells in the absence (Ctrl, grey histogram) or presence of inhibitors (blue histograms). Numbers of cells are indicated above bars. Dye uptake in the absence of BzATP is labelled -Bz. BzATP-induced dye uptake in non-transfected cells (NT) was normalized to another set of rP2X7-transfected control (*n* = 78 cells). Bars represent mean ± SEM (from 3 to 12 transfections). *p* values are from one-way Kruskal-Wallis ANOVA followed by Dunn’s comparison to respective controls. (**D**) Fluorescence spectra of 10 μM YO-PRO-1 bound to double strand DNA (dsDNA) recorded before (black trace, averaged from respective controls) and immediately after application of TA (red), 9-AC (blue) or FFA (green). Fluorescence spectra of 10 μM unbound YO-PRO-1 (in the absence of dsDNA) is indicated in grey. CaCC inhibitor concentrations were 20 μM TA, 1 mM 9-AC and 100 μM FFA.

**Figure 5 ijms-22-06542-f005:**
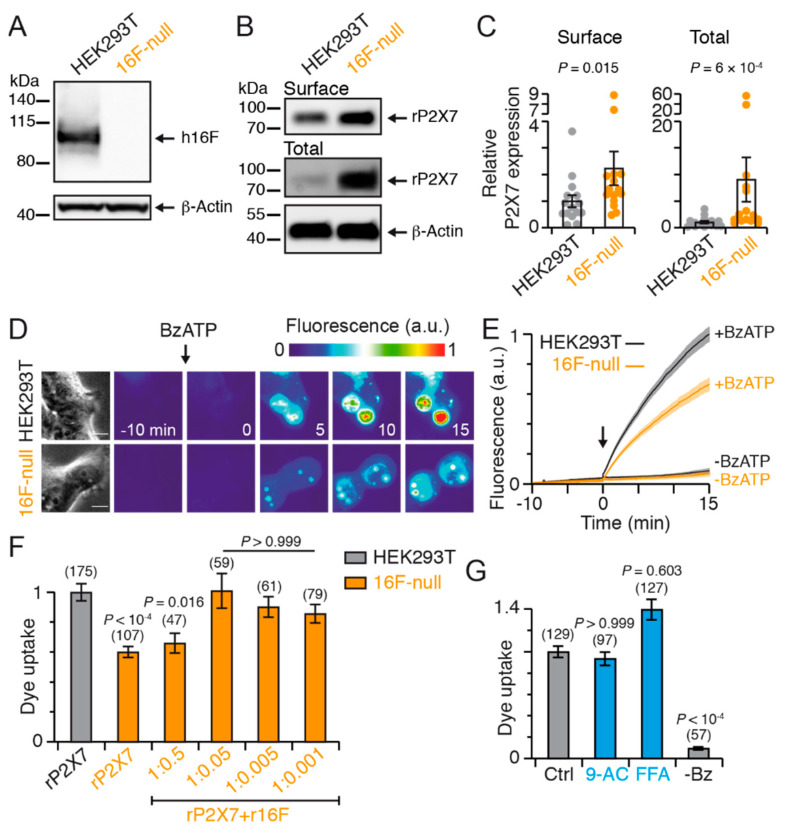
TMEM16F contributes to dye uptake. (**A**) Western blot analysis of total cell lysate separated by SDS-PAGE indicates the successful knock-out of endogenous hTMEM16F (h16F) by CRISPR/Cas9 in 16F-null cells. Western blotting was performed with anti-hTMEM16F antibody. β-Actin was used as a loading control. (**B**) Total and surface expression of myc tagged rP2X7 subunits transiently expressed in HEK293T or in 16F-null cells and resolved by SDS-PAGE. Proteins at the cell surface were biotinylated and isolated with neutravidin-bound agarose resin. Western blotting was performed with anti-c-myc antibody. β-Actin was used as a loading control. Apparent molecular weight markers are indicated. Uncropped blots are shown in Appendix A. (**C**) Summary of relative P2X7 expression normalized to β-actin. Bars represent mean ± SEM (from 15 transfections). *p* values are from Mann-Whitney test. (**D**) Time series fluorescence images of YO-PRO-1 dye uptake upon 10 μM BzATP activation (arrow) in HEK293T cells and 16F-null cells transiently expressing rP2X7. See legend of Figure 4A for details. Scale bar, 10 μm. (**E**) Corresponding YO-PRO-1 uptake over time in HEK293T cells (black) and 16F-null cells (orange), in the absence (-BzATP) or presence of 10 μM BzATP (+BzATP) applied as indicated (arrow). Solid lines represent mean ± SEM (shaded areas). (**F**) Summary of BzATP-induced YO-PRO-1 uptake in 16F-null cells transiently expressing rP2X7 (2 μg) in the absence (rP2X7) or presence of rTMEM16F (rP2X7 + r16F), transfected with varying quantities of cDNA (from 1 to 0.002 μg) at the indicated ratios (defined as rP2X7 to TMEM16F). Shown is the specific dye uptake, found by subtracting residual uptake recorded in the absence of BzATP. Data are normalized to rP2X7-expressing HEK293T cells. Bars represent mean ± SEM, and numbers of cells are indicated above bars (from 4 to 5 transfections). *p* values are from one-way Kruskal-Wallis ANOVA followed by Dunn’s comparison to HEK293T. (**G**) Summary of YO-PRO-1 uptake in 16F-null cells in the absence (Ctrl) or presence of TMEM16 inhibitors. Dye uptake in the absence of BzATP is labeled -Bz. Bars represent mean ± SEM, and numbers of cells are indicated above bars (from 5 transfections). *p* values are from one-way Kruskal-Wallis ANOVA followed by Dunn’s comparison to control (Ctrl). CaCC inhibitor concentrations were 1 mM 9-AC and 100 μM FFA.

**Figure 6 ijms-22-06542-f006:**
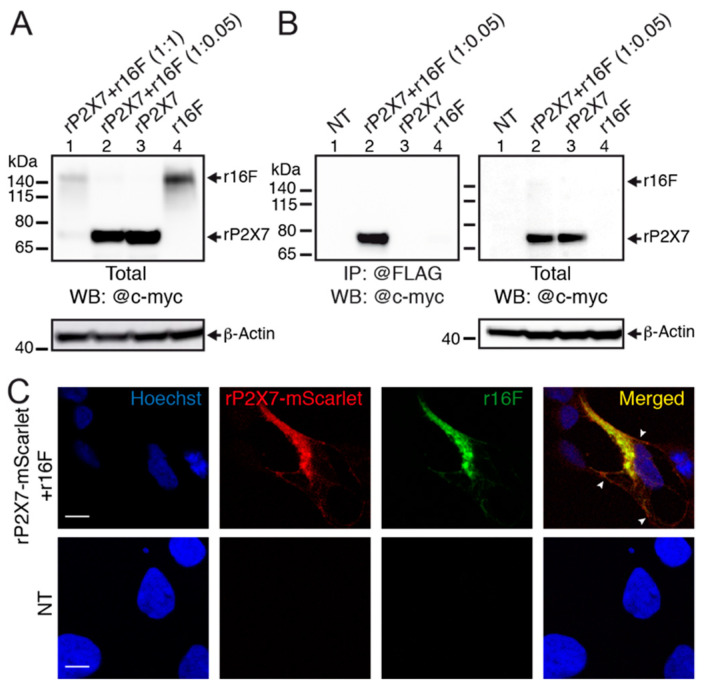
Physical proximity of P2X7 and TMEM16F. (**A**) Western blot of total cell lysate separated by SDS-PAGE from 16F-null cells transiently transfected with myc tagged rP2X7 (rP2X7) and myc-DDK tagged rTMEM16F (r16F) at a ratio of 1:1 (lane 1: 5 μg rP2X7 and 5 μg r16F) or 1:0.05 (lane 2: 5 μg rP2X7 and 0.25 μg r16F) and probed with anti-c-myc antibody. Note reduced rP2X7 subunit expression in lane 1 compared to that in lane 2. Control expressions of rP2X7 (10 μg) and r16F (10 μg) are shown in lane 3 and 4, respectively. β-Actin was used as a loading control. Apparent molecular weight markers are indicated. (**B**) Left: Co-immunoprecipitation reveals physical association of myc tagged rP2X7 and myc-DDK tagged r16F subunits transiently co-expressed at a ratio of 1:0.05 (5 μg rP2X7 and 0.25 μg r16F) in 16F-null cells and resolved by SDS-PAGE. Co-immunoprecipitation was performed in cell lysate with anti-FLAG antibody, which recognizes DDK tag of myc-DDK tagged r16F, and Western blotting was performed with anti-c-myc antibody. Right: Western blot of total cell lysate of corresponding samples probed with anti-c-myc antibody. Control expressions of rP2X7 (5 μg) and r16F (0.25 μg) are shown in lanes 3 and 4, respectively. Note that the band corresponding to myc-DDK tagged r16F is barely visualized because of the 20-fold lower expression of myc-DDK tagged r16F compared to that of rP2X7 (see, however, Appendix A for the actual presence of r16F). Controls in non-transfected (NT) cells are shown in lanes 1. β-Actin is used as the loading control. Uncropped blots are shown in Appendix A. (**C**) Top: Confocal images of 16F-null cells transiently transfected with 2 μg rP2X7-mScarlet (red) and 0.1 μg r16F (green) demonstrate co-localization (yellow) of both proteins. Head arrows indicate putative cell membrane localizations. Cells were stained with a primary anti-h16F antibody coupled to a secondary Alexa 488 conjugated antibody (green). Nuclei were stained with Hoechst (blue). Bottom: Control in non-transfected (NT) 16F-null cells. Scale bar, 10 μm.

**Figure 7 ijms-22-06542-f007:**
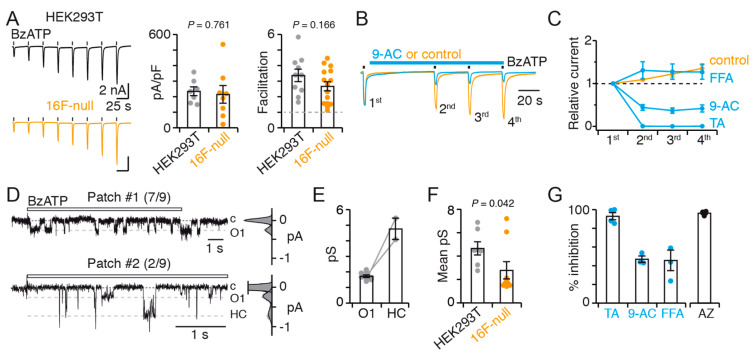
TMEM16F shapes P2X7 unitary conductance. (**A**) Left: whole-cell currents evoked by repeated 2 s applications of 10 μM BzATP recorded in HEK293T (black trace) and 16F-null cells (orange trace) transiently transfected with rP2X7. Middle: summary of current density of the 8th application (*n* = 7 for HEK293T cells and 8 for 16F-null cells). Bars represent mean ± SEM. Right: summary of current facilitation determined between the 8th and 1st application. See Materials and Methods for calculation (*n* = 10 cells for HEK293T and 15 for 16F-null). *p* values are from unpaired Student’s *t*-test. (**B**) Whole-cell currents evoked by repeated 2 s applications of 10 μM BzATP in the presence (blue) or absence of 9-AC (orange) in 16F-null cells transiently expressing rP2X7. Currents are normalized to the first application. (**C**) Summary currents relative to the first application in the absence (control) or presence of inhibitors (*n* = 4 cells for TA, 10 for 9-AC, 7 for FFA, and 8 for control). Symbols represent mean ± SEM. (**D**) Top (patch #1, representative of 7 out of 9 patches): single-channel currents elicited by 10 μM BzATP from an outside-out patch of 16F-null cell transiently transfected with rP2X7 exhibiting the O1 state (dashed grey line); c, closed state. Corresponding all points histogram of currents is shown right of the trace. Distributions were fit by the sum of two Gaussians. Bottom (patch #2, representative of 2 out of 9 patches): example of outside-out patches in which O1 and an additional high conductance (HC) were recorded (dashed grey lines). Currents were elicited by 10 μM BzATP. Corresponding all points histogram of currents is shown right of the trace. Distributions were fit by the sum of three Gaussians. Data were filtered at a final f_c_ of 100 Hz. (**E**) Summary of corresponding unitary conductances in 16F-null cells. Linking lines indicate data points that originate from the same patches (*n* = 9 patches). Bars represent mean ± SEM. (**F**) Summary of mean P2X7 unitary conductances in HEK293T and 16F-null cells. Mean was calculated by averaging the sum of each state conductance determined from each patch (*n* = 7 patches for HEK293T cells and 9 for 16-null cells). Bars represent mean ± SEM. *p* value is from Mann-Whitney test. (**G**) Summary of single-channel inhibition (*n* = 4 patches for TA, 3 for 9-AC, 3 for FFA, and 3 for AZ10606120). Bars represent mean ± SEM. Inhibitor concentrations were 20 μM TA, 1 mM 9-AC, 100 μM FFA and 1 μM AZ10606120. Single-channel and whole-cell data were recorded at −120 and −60 mV, respectively.

**Figure 8 ijms-22-06542-f008:**
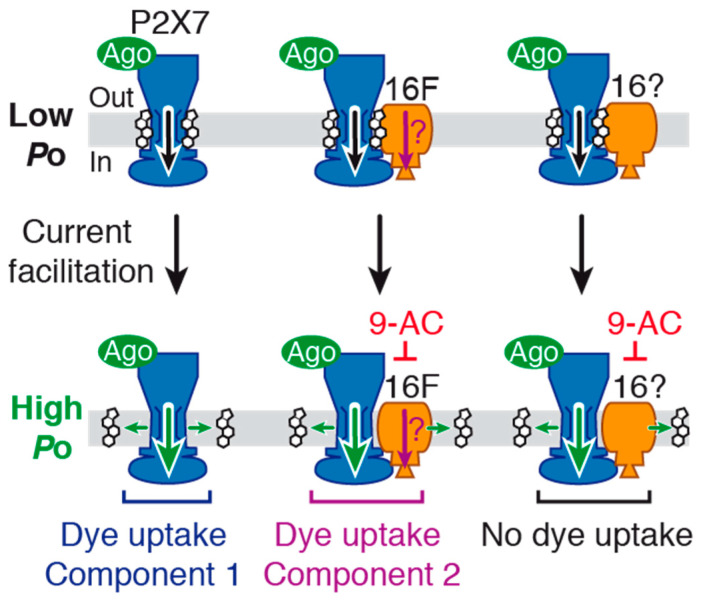
Proposed mechanisms underlying P2X7 current facilitation and macropore formation. P2X7 (blue) and TMEM16F (16F, orange) form functional complexes, in which both proteins are close to each other. In response to brief application of agonist (Ago, green spheres), cholesterol (hexagonal rings) maintains functional complex-embedded P2X7, and possibly “free” P2X7, in a low channel activity state (black arrows). Long or repeated agonist application transitions P2X7 channels from a low to high *P*o (enlarged green arrows), presumably through cholesterol dissociation (small lateral green arrows). Meanwhile, agonist binding to P2X7 activates at least two permeation components for large molecules, one that occurs directly through the P2X7 pore (component 1), and the other through functional coupling to TMEM16F (16F) (component 2). Note that the precise dye uptake pathway of component 2 (purple arrows) is unknown. Functional coupling with another TMEM16 subtype of unknown composition is also suggested. Membranes are represented as grey shadings. 9-AC (9-anthracene-carboxylic acid) is a non-selective TMEM16 inhibitor. See text for details.

## Data Availability

All data reported in this work are available upon request.

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
