# Peer review of "P2X7 Receptors and TMEM16 Channels Are Functionally Coupled with Implications for Macropore Formation and Current Facilitation"

_ijms, 2021, doi:10.3390/ijms22126542_

Round 1
Reviewer 1 Report
The authors investigated the mechanisms of macropore formation after activation of P2X7 receptors. They combined different approaches to resolve this issue. They found that P2X7 and TMEM16 channels form functional complexes in both HEK293T and Xenopus oocytes. They reported three mayor findings: First, TMEM16F influences P2X7 expression and function. Second, the functional interplay between P2X7 and other members of the TMEM16 family and third, the large permeability proceeds through at least two components: a direct passage through P2X7 pore and an alternative permeation through TMEM16F.
This is a clear written manuscript und the data are convincing. One disadvantage is that overexpressing systems (in particular rat P2Y7) are used and the data are not confirmed for example in immune cells, which express the investigated proteins endogenously. Functional interactions of proteins occur often in specific micro-domains. Proteins which are overexpressed may be misplaced and show altered behaviour. This circumstance should be discussed by the authors.
Minor points:
Figure 8: Explanation of Ago and 9-AC in the legend is missing.
Line 849: concentration of @M
Author Response
We would like first to thank the reviewer for her/his very helpful comments. A point-by-point response can be found below.
This is a clear written manuscript und the data are convincing. One disadvantage is that overexpressing systems (in particular rat P2Y7) are used and the data are not confirmed for example in immune cells, which express the investigated proteins endogenously. Functional interactions of proteins occur often in specific micro-domains. Proteins which are overexpressed may be misplaced and show altered behaviour. This circumstance should be discussed by the authors.
Response: We agree with this statement, and accordingly we added in the discussion (line 555) the following sentences: “In addition, because all experiments described in this study were carried out in overexpressing systems, overexpressed P2X7 and TMEM16F may be misplaced in terms of cellular localization, and show altered behavior that may not be exactly representative of the native interaction. Therefore, further work is also needed to demonstrate that this functional interaction is also relevant to endogenous proteins.”
Minor points:
Figure 8: Explanation of Ago and 9-AC in the legend is missing.
Response: We provide explanation in the legend
Line 849: concentration of @M
Response: We fixed it.
Reviewer 2 Report
This is overall a very good paper. The study is well designed, the methods are described in detail, research results are carefully analyzed and discussed. My only concern is about the explanation of the involvement of second component of the P2X7/TMEM16F complex. Is there any possibility of the engagement of the other component that is not Ca2+ - activated Cl- channel?
Author Response
We would like first to thank the reviewer for her/his very helpful comments. A point-by-point response can be found below.
My only concern is about the explanation of the involvement of second component of the P2X7/TMEM16F complex. Is there any possibility of the engagement of the other component that is not Ca2+ - activated Cl- channel?
Response: We cannot formally exclude the possibility of another component that is not a Ca2+ - activated Cl- channel. However, this seems unlikely, because we provide genetic evidence (16F-null cell line) that TMEM16F contributes to this second component. Evidence is provided for this in the text. That said, it remains possible that other components that may not incorporate a Ca2+ - activated Cl- channel may contribute to P2X7 function, but it is highly speculative, and we provide no evidence for this.